# The Future of Targeted Therapy for Leiomyosarcoma

**DOI:** 10.3390/cancers16050938

**Published:** 2024-02-26

**Authors:** Ryan A. Denu, Amanda M. Dann, Emily Z. Keung, Michael S. Nakazawa, Elise F. Nassif Haddad

**Affiliations:** 1Division of Cancer Medicine, The University of Texas MD Anderson Cancer Center, Houston, TX 77030, USA; radenu@mdanderson.org; 2Division of Surgical Oncology, Department of Surgery, The University of Texas Southwestern Medical Center, Dallas, TX 75390, USA; amanda.dann@utsouthwestern.edu; 3Department of Surgical Oncology, The University of Texas MD Anderson Cancer Center, Houston, TX 77030, USA; ekeung@mdanderson.org; 4Department of Sarcoma Medical Oncology, The University of Texas MD Anderson Cancer Center, Houston, TX 77030, USA

**Keywords:** leiomyosarcoma, genomics, transcriptomics, immunotherapy, precision oncology, targeted therapy

## Abstract

**Simple Summary:**

Leiomyosarcoma is a subtype of soft tissue sarcoma with poor outcomes and response to currently available treatments. The goal of this review is to assess the current landscape of targeted therapies and explore how our current understanding of the biology of leiomyosarcoma may inform the development of new targeted therapies to improve the treatment of leiomyosarcoma.

**Abstract:**

Leiomyosarcoma (LMS) is an aggressive subtype of soft tissue sarcoma that arises from smooth muscle cells, most commonly in the uterus and retroperitoneum. LMS is a heterogeneous disease with diverse clinical and molecular characteristics that have yet to be fully understood. Molecular profiling has uncovered possible targets amenable to treatment, though this has yet to translate into approved targeted therapies in LMS. This review will explore historic and recent findings from molecular profiling, highlight promising avenues of current investigation, and suggest possible future strategies to move toward the goal of molecularly matched treatment of LMS. We focus on targeting the DNA damage response, the macrophage-rich micro-environment, the PI3K/mTOR pathway, epigenetic regulators, and telomere biology.

## 1. Introduction

Leiomyosarcoma (LMS) is an aggressive form of soft tissue sarcoma derived from smooth muscle cells. Most commonly arising in the uterus (ULMS) and retroperitoneum (often from large blood vessels such as the inferior vena cava), LMS can also be seen in the extremities and in skin. Non-uterine LMS will be hereafter referred to as soft tissue LMS (STLMS) and will exclude cutaneous LMS [1]. LMS represents 10–20% of all soft tissue sarcomas and is characterized by high risk of distant recurrence, with rates of distant metastasis at 10 years ranging from 31 to 71% depending on the site of origin [2,3,4,5]. The five-year survival rate is 42% for all stages combined. Its rarity and heterogeneity have contributed to our limited understanding of LMS biology, and ongoing work seeks to better understand the molecular underpinnings of this disease, identify therapeutic targets, and improve patient outcomes.

Surgery is the mainstay of treatment for localized LMS, with the addition of radiation and chemotherapy depending on the clinical characteristics and associated local and distant recurrence risks [6]. Treatment of recurrent or metastatic LMS typically involves cytotoxic chemotherapy. The LMS-04 trial demonstrated the efficacy of doxorubicin plus trabectedin over doxorubicin alone with a response rate of 36% compared to 13%, respectively, and a disease control rate (response or stable disease) of 91.9% versus 78.9% [7]. This represents the most efficacious chemotherapy regimen prospectively validated to date for LMS and the first trial to demonstrate an overall survival (OS) benefit with an anthracycline-based chemotherapy combination in the first line setting for advanced LMS. Alternative treatments include other doxorubicin-based doublet regimens (doxorubicin plus ifosfamide, doxorubicin plus dacarbazine) [8] with response rates of around 30% but with no prospective randomized control data showing an OS benefit, and gemcitabine plus docetaxel with response rates of about 20% [9,10], or single agent chemotherapies (e.g., doxorubicin, gemcitabine, docetaxel) with response rates of about 10% [11,12]. Therefore, there is great unmet clinical need to develop better treatments for LMS.

## 2. Genomic Landscape of Leiomyosarcoma

Sarcomas can be divided into two main categories based on their genomic features: (1) translocation- or fusion-driven and (2) complex karyotypes. Based on a number of studies assessing the genomic landscape, LMS falls under the latter category of complex karyotypes [13,14,15]. Several studies have assessed the genomic landscape of LMS. The studies with next-generation sequencing data are summarized in Table 1. In general, the LMS genome lacks targetable alterations with the currently available therapeutics.

Initial genomic studies involved array comparative genomic hybridization (aCGH) analyses to identify copy number variants (CNVs). These aCGH studies identified recurrent losses in 10q, 11q, 13q, and 2p and gains in Xp, 5p, 8q, and 17p [25,26,27,28,29]. A loss of 10q and a gain of 5p were associated with higher grade, larger tumor size, and higher rates of metastasis [25]. Chromosomes 10q and 13q encompass *PTEN* and *RB1*, respectively, suggesting an important role for these pathways in LMS biology [15].

The advent of next-generation sequencing significantly advanced our understanding of the LMS genome. The Cancer Genome Atlas (TCGA) initiative performed a multi-omic analysis of 206 sarcomas including 80 LMS (53 STLMS and 27 ULMS) [16]. Overall, sarcomas showed a lower total mutational burden (TMB) compared to other cancer types. Further, sarcomas showed more CNVs than most cancer types, though LMS generally had less CNVs than other complex-karyotype sarcomas. Recurrent mutations in LMS were seen in *TP53*, *ATRX*, and *RB1* [16]. Additionally, *MYOCD* amplification, *PTEN* deletion/mutation, and activation of AKT/mTOR pathways were enriched compared to other sarcoma subtypes. This analysis also highlighted the genomic differences between STLMS and ULMS, which have distinct clinical behaviors [30]. Although STLMS and ULMS are similar with respect to CNVs, they differ significantly in methylation profiles and mRNA expression signatures. ULMS was associated with a DNA damage response (DDR) signature, while STLMS showed a HIF-1α signature.

Additional sequencing studies have also confirmed recurrent alterations in *TP53*, *RB1*, and *ATRX* in LMS [17,20,22]. Biallelic disruption of *TP53* and *RB1* occur in 92% and 94% of LMS cases as a consequence of any of a number of observed genomic alterations [20], suggesting that loss of *TP53* and *RB1* is essentially universal in both STLMS and ULMS. *MED12* mutations also occur at high frequency but only in ULMS [21,22]. CNV analysis from these studies have also validated some of the initial findings from aCGH studies, showing recurrent losses of chromosomal regions involving key tumor suppressor genes such as *PTEN* (10q), *RB1* (13q), *CDH1* (16q), and *TP53* (17p), while the most frequent copy number gains involved chromosome regions 17p11.2 (*MYOCD*) and 15q25-26 (*IGF1R*) [20,22].

Genomic studies have shown evidence of chromothripsis, a massive rearrangement of the genome that occurs during a single catastrophic event [31], in 35% of LMS [20]. Therefore, a subset of the complex genomes observed in LMS may be attributable to chromothripsis. Further, whole-genome duplication resulting in polyploidy occurs in 55.1% of LMS [20]. Interestingly, data from paired primary tumor and metastasis samples show whole-genome doubling in the metastases but not the primary tumor, suggesting that genome doubling may contribute to progression and metastasis [20]. Only mutant *TP53* and *RB1* were detected in the metastatic tetraploid samples, suggesting that loss of *TP53* and *RB1* wild-type allele in the primary tumor may be a precursor event to metastatic development [20]. Concordantly, tetraploidy was shown to induce a *TP53*-dependent cell cycle arrest; therefore, loss of *TP53* is likely required for proliferation of tetraploid cells [32]. Furthermore, tetraploidy is likely an intermediate phenotype that is permissive of losses of whole chromosomes, leading to evolution of the cancer genome, including losses of chromosomes with critical tumor suppressors and gain of chromosomes with oncogenes, and potentiating progression [33,34].

Recent large scale clinical targeted next-generation sequencing testing from Foundation One (1493 LMS cases) [35], Caris (751 LMS cases) [18], and Memorial Sloan Kettering IMPACT (290 LMS cases) [36], have largely found similar genomic alterations. TP53 pathway alterations were seen in 42–64% STLMS and 41–68% of ULMS [18,36]. The incidence of TP53 pathway alterations is lower in these targeted gene panels than was reported in some of the aforementioned studies, perhaps due to lower sensitivity with targeted gene panels. The PI3K pathway was altered in 20% of STLMS and 30% of ULMS, and *PTEN* loss was the most common pathway alteration [36]. Additionally, gene alterations in the DDR pathway were seen in 10% of STLMS and 24% of ULMS, the epigenetic pathway in 27% of STLMS and 49% of ULMS, and cell cycle in 48% of STLMS and 60% of ULMS [36]. Actionable fusions were seen in 1.9% of LMS; most of these were *TNS1–ALK* fusions [35], which are predicted to be sensitive to currently available ALK inhibitors such as alectinib and brigatinib.

Data from our analysis of the AACR GENIE database are largely consistent with the above findings. This dataset includes 723 STLMS and 374 ULMS. The most commonly mutated gene in both STLMS and ULMS was *TP53*, with mutations in 49.5% and 53.7%, respectively (Figure 1A). *ATRX* mutations were more prevalent in ULMS than STLMS. *RB1* mutations were also prevalent in both STLMS and ULMS. The most common CNV was *RB1* deletion, which was more prevalent in ULMS (39.8% versus 24.8% in STLMS; Figure 1B). Overall, TMB was low (Figure 1C; median 2.56 mutations/megabase). High TMB (>10 mutations/megabase) was seen in 9.7% of STLMS and 6.5% of ULMS compared to 5.6% in all sarcomas in the GENIE database (Figure 1C). Taking all these alterations together and assessing for clinical actionability using OncoKB, a precision oncology knowledge base that reports recommendations of different levels of strength based on the evidence reported in the literature, we find that 32.6% of STLMS and 43.6% of ULMS have potentially actionable alterations (Figure 1D).

## 3. Transcriptomic Profiling Has Revealed Distinct Molecular Subtypes of Leiomyosarcoma

The transcriptomic landscape was also assessed in several studies (summarized in Table 2), and multiple studies have independently identified three distinct molecular subgroups of LMS from these data. One of the first studies profiled 40 LMS cases, finding that 11 of these clustered together and noted enrichment in muscle-associated genes, while the remaining 29 were more heterogeneous [39]. A subsequent study identified three molecular subgroups after performing gene expression analysis of 51 LMS tumors using microarrays [40]. Group I consisted mostly of STLMS and was defined by muscle-associated gene enrichment and more CNV, including losses of 1p36.32 and 16q24. Group II was also composed mostly of STLMS and was defined by enrichment in genes related to protein metabolism and regulation of cell proliferation. Group III comprised approximately equal numbers of STLMS and ULMS and was enriched in genes related to organ and system development, metal binding, extracellular proteins, wound response, and ribosomal proteins. The majority of these samples were treatment-naïve (e.g., no exposure to neoadjuvant radiation or chemotherapy), and there were no differences in the distribution of previously treated tumors between groups. There was also no difference in patient age or tumor grade between the subtypes. Group I was reproduced using gene expression data from another LMS gene expression profiling study, though Groups II and III could not be reproduced [41]. This same group later validated their findings using a larger cohort of 99 LMS cases from different institutions and profiling by 3′ end RNA sequencing [42]. This analysis again identified three subtypes. They identified a panel of immunohistochemistry (IHC)-based markers, including ACTG2, SLMAP, LMOD1, CFL2, MYLK, and ARL4C, to distinguish the three subtypes and subsequently correlated these with clinical outcomes in a separate cohort using a clinically annotated tissue microarray [42]. This showed that Subtype I was associated with better disease-specific survival (DSS), Subtype II was associated with worse DSS, and Subtype III was associated with intermediate outcomes [42].

The next study assessed 49 LMS tumors using RNA sequencing and again identified three LMS subgroups [20]. Subgroup 1 was defined by expression of *LMOD1* and smooth muscle markers, greater differentiation, and enrichment in platelet degranulation, complement activation, and metabolism gene signatures. Subgroup 2 was defined by *ARL4C* expression, dedifferentiation, and enrichment in muscle development and function and regulation of membrane potential. Subgroup 3 was mostly ULMS and was defined by myofibril assembly, muscle filament action, and cell–cell signaling gene signatures. Subgroups 2 and 3 in this study were similar to the previously identified Subtypes II and I, respectively, from the previous study [42], based on the elevated expression of *ARL4C* or *CASQ2* and *LMOD1*, respectively.

Next, the TCGA initiative analyzed bulk RNA sequencing and also identified the following three molecular subtypes of LMS with distinct transcriptomic profiles and clinicopathological characteristics: (1) uterine subtype associated with poor prognosis; (2) soft tissue C1, characterized by 17p11.2 deletion and DNA hypermethylation, and associated with poor prognosis; and (3) soft tissue C2 characterized by an inflammation gene expression signature and lower levels of DNA methylation, and associated with better prognosis [16].

Lastly, a study from the Toronto group performed bulk RNA sequencing of 51 LMS tumors and integrated it with 79 transcriptomes from TCGA [24]. This analysis also identified three distinct subtypes. Subtype 1 contained LMS from different anatomic sites, harbored higher TMB, and was associated with worse OS and DSS. Subtype 2 was mostly abdominal and was associated with better OS and DSS compared to the other subtypes. Subtype 3 was mostly uterine, harbored higher TMB, and was associated with worse OS and DSS [24].

Currently, integration of these molecular subtypes into widespread clinical practice has not been realized. One challenge has been identifying the best markers to use to ascribe tumors to a LMS subtype. While each of the aforementioned studies describe three different LMS molecular subtypes, the genes defining these subtypes do not fully overlap across the studies. These differences will need to be reconciled for the field to move forward and be able to meaningfully utilize these subtypes in clinical practice. One such exciting potential application of LMS molecular subtypes is in the selection and stratification of patients for clinical trials. Currently, patients with LMS are generally enrolled together on the same trials, regardless of anatomic site or genomic differences, although this may not be appropriate if the biology and clinical outcomes differ between the different subtypes. It will be interesting to see how future clinical trials integrate these molecular subtypes prospectively to develop more personalized therapies for different LMS subtypes.

## 4. Approved Targeted Therapies for Leiomyosarcoma

There are no FDA-approved targeted therapies specifically for LMS. Pazopanib, a tyrosine kinase inhibitor with high affinity for VEGFR, PDGFR, KIT, and FGFR [43], is FDA-approved for advanced soft tissue sarcomas, including LMS, that have received prior chemotherapy. This is based on the results of the PALETTE trial, which showed that pazopanib significantly prolonged median progression-free survival (PFS, 4.6 vs. 1.6 months for placebo), and a clinical benefit was seen in 73% (6% partial response and 67% stable disease) with pazopanib versus 38% (stable disease only) with placebo [44]. Specific to ULMS, pazopanib showed a median PFS of 3.0 months, OS of 17.5 months, and objective response rate (ORR) of 11% [45]. Other than pazopanib, the only approved targeted therapies for LMS are those with tissue-agnostic approvals. These include pembrolizumab and dostarlimab for microsatellite instability-high or TMB-high tumors [46,47], BRAF/MEK inhibitors dabrafenib plus trametinib for *BRAF* V600E mutant tumors, the *RET* inhibitor selpercatinib for cancers with *RET* fusions [48], and *NTRK* tyrosine kinase inhibitors (TKIs) such as entrectinib and larotrectinib for *NTRK* fusions [49]. However, as described above, these alterations are rarely seen in LMS. Based on data in the GENIE database, 9.7% of STLMS and 6.5% of ULMS have high TMB (Figure 1). Regarding other OncoKB level 1 alterations (FDA-recognized biomarker predictive of response to an FDA-approved drug), there was only one ULMS case of *BRAF* V600E, one ULMS case with *RET* fusion, and one ULMS case with *NTRK* fusion (Appendix A). Regarding level 2 alterations (standard of care biomarker recommended by NCCN predictive of response to an FDA-approved drug), the most common was the *BRCA2* mutation, seen in seven cases of ULMS.

Hormone therapy targeting the estrogen and progesterone receptors (ER and PR) was also utilized to treat LMS, particularly ULMS [50,51,52]. These were based on findings that 25–60% and 35–65% of ULMS express ER and PR, respectively [53,54,55,56,57]. Higher ER and PR expression were associated with better survival [58]. A single-arm phase 2 trial of the aromatase inhibitor letrozole treated 27 patients with advanced ULMS and demonstrated a 12-week PFS of 50%. The best response was the stable disease in 54%, and the median duration of treatment was 2.2 months [59]. However, most other evidence comes from retrospective analyses or case reports [58]. For example, a retrospective study of 16 metastatic ER-positive ULMS patients treated with aromatase inhibitors found clinical benefit (complete response, partial response, or stable disease for at least 6 months) in 10 patients, and the benefit was greater in low grade compared to high grade (mPFS 20 months versus 11 months) [50]. However, in general, the success of these hormone therapies has been limited, the quality of data supporting them is low, and these agents are falling out of favor, particularly at sarcoma centers. Nevertheless, there may be some role for these treatments in low-grade ULMS, as a retrospective study of 27 patients demonstrated a 52% partial response and 37% with a stable disease, and mPFS was not reached [60].

## 5. Pharmacogenomic Biomarkers in Leiomyosarcoma

As mentioned, the treatment of recurrent or metastatic LMS typically involves cytotoxic chemotherapy. However, there are no existing biomarkers readily available in clinical practice that predict the response to chemotherapy in LMS. Given the modest response rates with combination chemotherapy (20–30%) relative to the toxicity (for example grade 3–4 neutropenia of 43% with doxorubicin plus ifosfamide) [8], it would be ideal to have a predictive biomarker to avoid unnecessary toxicity in patients unlikely to benefit from chemotherapy.

Patient differences in the pharmacologic response to chemotherapy is a common cause of patient morbidity. Pharmacogenomic biomarkers are typically genetic variants in metabolism enzymes, membrane transporters, or drug targets and can predict drug response. Some of these biomarkers exist for LMS. For example, increased expression of the nucleoside transporter ENT1 was associated with an increased response to gemcitabine in LMS [61]. Further, low gene expression of *BRCA1* and increased gene expression of *XPG* were associated with a better response to trabectedin in advanced sarcomas, and this cohort of 245 patients included 17.5% with STLMS and 7.5% with ULMS [62]. Additional work is needed to identify and implement pharmacogenomic biomarkers into current clinical practice in LMS, as well as undertake these types of biomarker analyses in future drug development studies.

## 6. The DNA Damage Response in Leiomyosarcoma

The DDR is a complex coordinated network of pathways designed to repair diverse insults to our DNA. DNA damage is critical for mutagenesis, which drives the development of most cancers. These pathways have been exploited to treat many cancers. Homologous recombination (HR) is one type of DDR, which repairs double-strand DNA breaks. It is so named because it uses a homologous DNA segment as a template for DNA repair, thereby allowing for faithful repair of double-strand DNA breaks. The *BRCA1* and *BRCA2* genes are integral for HR, and *BRCA1/2* mutations are associated with increased double-strand breaks [63]. Individuals with germline mutations in *BRCA1/2* are predisposed to developing multiple different types of cancer, namely breast, ovarian, and prostate [64]. Alterations in HR have significant therapeutic relevance. Notably, synthetic lethality was demonstrated with *BRCA1/2* alterations in ovarian cancer and inhibition of poly (ADP-ribose) polymerase 1 (*PARP1*; Figure 2) [65,66,67]. PARP1 is involved in repairing single-strand break repairs, and the initial mechanism of synthetic lethality between PARP1 inhibitors and HR deficiency was thought to be that PARP1 inhibition resulted in more single-strand breaks that led to replication-associated double-strand breaks, though in vitro evidence for this is lacking. The true mechanism likely involves PARP1 accumulating at stalled replication forks and becoming hyperactivated in HR-deficient cells [68]. This finding of synthetic lethality has led to the use of PARP inhibitors such as olaparib, rucaparib, niraparib, and talazoparib in the treatment of tumors with alterations in *BRCA1/2* mutations and other HR pathway genes (Figure 2). Further, HR mutations were associated with increased platinum sensitivity in other malignancies such as pancreatic and urothelial cancers [69,70]. In LMS, *BRCA1/2* alterations are seen more often in ULMS, with reported incidences of 10–22% of ULMS versus 1–11% of STLMS [18,71]. Furthermore, data from IMPACT tumor profiling showed that 20% of LMS have an oncogenic DDR gene alteration, including 25% of ULMS and 14% of STLMS, and 18% of ULMS and 10% of STLMS harbored an alteration specifically in an HR gene [72]. The most frequently altered DDR genes were *BRCA2* (7%), *RAD51B* (4%), and *ERCC5* (2%). These patients with HR-deficient tumors, especially non-*BRCA1/2* mutant tumors, had worse outcomes compared to patients without DDR gene alterations [72]. A different study found higher rates of deleterious mutations in HR components, including *BRCA2* (53%), *ATM* (22%), *CHEK1* (22%), *XRCC3* (18%), *CHEK2* (12%), *BRCA1* (10%), and *RAD51* (10%) [20]. In another cohort of 58 ULMS assessed by whole-exome or whole-genome sequencing for HR deficiency, five (8.6%) had HR deficiency, and all five were treated with PARP inhibitors [73]. Two of three patients with mature clinical follow up achieved a complete response or durable partial response with the subsequent addition of platinum to PARP inhibitor upon minor progression after an initial PR on PARP inhibitor. Another study reported *BRCA1* mutations in 0 of the 41 LMS cases analyzed, *BRCA2* mutations in 11.1% of STLMS, and 23% of ULMS [18]. Lastly, data from the GENIE dataset identified *BRCA2* mutations in 2.2% of LMS (Appendix A); no other actionable alterations in HR genes were seen in this cohort.

*ATM* is a kinase that is also involved in repairing double-strand breaks by the phosphorylation of several substrates including BRCA, CHEK1, CHEK2, and TP53. *ATM* alterations were reported in 16% of LMS [22], and loss of ATM expression by immunohistochemistry (IHC) was reported in 55% of LMS cases [73]. Another study reported *ATM* mutations in 3.9% of STLMS and 2.4% of ULMS [18]. In the GENIE database, 1.4% of STLMS and 2.1% of ULMS have pathogenic and potentially actionable *ATM* mutations (Appendix A). Preclinical studies have shown that DDR alterations, such as *ATM* loss-of-function, can sensitize tumors to inhibitors of ATR, another kinase involved in DDR, primarily of single-stranded break repair [74]. Current clinical trials are underway with ATR inhibitors in biomarker-selected patients; however, this has not yet been explored in LMS.

Mismatch repair (MMR) is a type of DDR that targets base–base mismatches and insertion/deletion loops [75]. Cells deficient in mismatch repair are unable to fix these mismatches and acquire spontaneous mutations at a much higher rate. Germline alterations in mismatch repair genes (*MLH1*, *MSH2*, *MSH6*, and PMS2) cause Lynch syndrome. Generally sarcomas are not associated with Lynch syndrome, though there have been few cases of Lynch-associated LMS in the literature [76]. In one study looking at mismatch repair deficiency (dMMR) or microsatellite instability (MSI), only 1 of 25 ULMS and 0 of 40 STLMS exhibited dMMR or MSI [77].

There were clinical trials that have attempted to exploit vulnerabilities in DDR in LMS. The phase 1 TOMAS trial investigated trabectedin plus the PARP inhibitor olaparib in advanced sarcomas. Of the 15 patients with LMS in this trial, five patients had prolonged clinical benefit of over 5 months [78]. The TOMAS2 trial further investigated this combination versus trabectedin alone in advanced sarcomas, and 29 patients with LMS (15 STLMS, 14 ULMS) were enrolled [79]. The mPFS was 3.9 vs. 2.9 months, though not statistically significantly different, 20% treated with the combination were treated for over a year. Another example is the NCI protocol 10250, a recent phase 2 trial of olaparib in combination with temozolomide in advanced ULMS, which enrolled 22 pre-treated patients and showed an ORR of 27% and mPFS of 6.9 months [80]. The benefit appeared to be greater for patients with tumors harboring deleterious HR gene alterations.

Preclinical data and early-phase clinical trials suggested that PARP inhibitors have synergistic activity when administered in combination immunotherapy [81]. This was tested in the JAVELIN PARP phase II trial combining avelumab with talazoparib, which showed higher response rates in *BRCA*-altered than non-*BRCA*-altered tumors, though no sarcomas were included in this trial [82]. Specific to sarcoma, the DAPPER trial assessed the anti-PDL1 antibody durvalumab in combination with either olaparib or cediranib (VEGF TKI), with 25 patients in the LMS cohort [83]. No responses were seen, though seven (30.4% of evaluable patients) had stable disease, and mPFS was 9 and 4 months for the olaparib and the cediranib groups, respectively. It will be interesting to assess novel DDR pathway inhibitors in LMS, both alone and in rational combinations, such as with immunotherapy.

## 7. The PI3K/PTEN/AKT/mTOR Signaling Pathway

The PI3K pathway is one of the most significantly and frequently disordered pathways in cancer. The PI3K/PTEN/AKT/mTOR pathway is a complicated cascade of phosphorylation and dephosphorylation by kinases and phosphatases. Activation of the pathway starts with engaging a cell surface receptor, such as extracellular growth factor binding a receptor tyrosine kinase (Figure 3A). Pathway activation can also result from mutations in the receptor or any downstream signaling components. PI3K is a family of lipid kinases that phosphorylate the phospholipids. Upon activation and autophosphorylation of a receptor tyrosine kinase on the cell membrane, PI3K is recruited to the membrane and binds the phosphotyrosine residue on the growth factor receptor. This results in allosteric activation of PI3K, resulting in phosphorylation of nearby phospholipid and catalyzes PIP2 to PIP3. This recruits and activates AKT, which phosphorylates a myriad of targets involved in metabolism, cell cycle regulation, and inhibition of apoptosis [84]. *PTEN* is a phosphatase that negatively regulates the pathway by dephosphorylation of PIP3, the PI3K target (Figure 3A). *PTEN* was shown to act as a tumor suppressor in many different contexts and inhibits cell growth and increases sensitivity to apoptosis [85]. The mTOR complex 1 (mTORC1) regulates the assembly of the eukaryotic initiation factor 4F complex (eIF4F), which drives translation of mRNAs that are important for cell proliferation. mTORC1 is a target of rapamycin and serine/threonine kinase that is ubiquitously expressed in human cells. It plays an important role in transducing the signaling at the membrane to downstream targets, resulting in enhanced metabolism, autophagy, invasion, and metastasis [86].

As previously mentioned, genomic profiling studies in LMS have demonstrated recurrent losses of chromosome 10 region encompassing *PTEN* [87,88]. Data from the GENIE dataset identified PTEN alterations in 10% of STLMS and 16% of ULMS, and most of these alterations are deletions (Appendix A). Other studies have also shown *PTEN* alterations in 4.4–75% of LMS, depending on the type of alteration and analysis [16,17,18,19,89]. Furthermore, activation of the PI3K/AKT/mTOR pathway was shown in LMS samples by elevated levels of phosphorylated AKT, and deletion of *Pten* in smooth muscle lineage induces smooth muscle cell hyperplasia and LMS in mice [90].

PI3K/AKT/mTOR pathway activity was extensively reported as a driver of therapy resistance in multiple cancers and different treatment contexts [65]. *PTEN* loss was associated with resistance to TKIs in gastrointestinal stromal tumors [91]. *PTEN* loss was associated with increases in expression of immunosuppressive cytokines and resistance to immunotherapy, including in ULMS [92]. Treatment with a selective inhibitor of PI3Kβ, a PI3K isoform that can activate AKT activity in cancer cells but is dispensable for T-cell receptor activation and signaling, improved the efficacy of both anti-PD1 and anti-CTLA4 in a model of *PTEN*-depleted melanoma cells [93]. Additionally, *PTEN* loss inhibits autophagy to reduce T-cell-mediated killing, and increasing expression of autophagy genes restored sensitivity to T-cell-mediated killing [93]. As the role of autophagy and autophagy inhibitors is complicated, controversial, and likely context-dependent, it will be important to validate these findings in LMS models. In summary, loss of *PTEN* may be a mechanism of resistance to multiple classes of therapy in LMS, and inhibition of this pathway warrants further exploration.

Inhibitors of this PI3K/AKT/mTOR pathway were approved for cancer treatment. Specifically, rapamycin derivatives such as everolimus are used namely in breast cancer and neuroendocrine tumors [94]. However, in LMS, these inhibitors remain an active area of investigation. A phase 3 study of the mTOR inhibitor ridafarolimus versus placebo in 702 heavily pre-treated advanced bone and soft tissue sarcomas showed a clinical benefit rate (complete response, partial response, or stable disease for at least 16 weeks) of 40.6% versus 28.6%, and mPFS was 17.7 weeks versus 14.6 weeks with placebo [95]. The LMS cohort compared similarly to the other subtypes in the study. One possible reason for the limited success seen with mTOR inhibitors is that such inhibition creates a negative feedback loop mediated by mTORC2 [96]. In human LMS cell lines, use of a dual PI3K and mTOR inhibitor was found to suppress this negative feedback loop but also increase ERK pathway activity [97]. The addition of a MEK inhibitor synergized with PI3K/mTOR inhibition [97], suggesting this combination as a possible strategy (Figure 3B).

There are other potential ways to block this pathway in LMS. First, inhibition of the PI3Kβ isoform may inhibit PI3K signaling in tumor cells while preserving anti-tumor immunity [93]. Second, *Nemo-like kinase* (NLK), *Polo-like kinase 4* (*PLK4*) and *TTK* (alias MPS1) were identified in synthetic lethal screens in *PTEN*-deficient cancer cells and may warrant further investigation in LMS [98].

## 8. Targeting the Micro-Environment

The LMS micro-environment largely consists of macrophages and T-cells [99]. Tumor-associated macrophages generally fall on a spectrum between pro-inflammatory M1 macrophages and immunosuppressive M2 macrophages. M1 macrophages preferentially promote the Th1 effector response and express high levels of TNF, iNOS, and MHC class II. They can phagocytose and destroy cancer cells in addition to bacteria and other microbes. M2 macrophages are driven by Th2 cytokines such as IL4 and IL13 and express arginase-1 (ARG1), IL-10, CD163, CD204, or CD206 [100]. The majority of macrophages in the LMS micro-environment have the M2 phenotype, as evidenced by the majority of CD14+ cells co-expressing the M2 marker CD163 in IHC analysis of LMS, and 61% of LMS exhibit greater than 20% infiltration by these M2 macrophages [99]. This is largely preserved in the LMS metastatic tumor micro-environment [99]. Clinically, greater M2 macrophage infiltration was associated with worse OS and DSS in LMS [99,101].

Response to immunotherapy has been limited to date in LMS, and the role of immunotherapy needs to be better defined for LMS. One of the challenges is the small number of patients included in single arm studies of various combinations, making it hard to delineate a specific strategy of interest or a predictive biomarker/molecular subgroup. A phase 2 trial of the anti-PD1 antibody nivolumab in 12 patients with advanced ULMS had no responses, and mPFS was 1.8 months [102]. The SARC028 trial evaluated another anti-PD1 agent, pembrolizumab, in advanced sarcomas. Ten patients with LMS were enrolled, and again no clinical responses were observed [103]. In the Alliance A091401 trial investigating ipilimumab plus nivolumab versus nivolumab monotherapy, two of fourteen LMS patients in the ipilimumab plus nivolumab arm (one STLMS, one ULMS) and one of fifteen LMS patients in the nivolumab arm (STLMS) had responses [104]. Next-generation immune checkpoint inhibitors are also being evaluated. For example, the C-800-01 trial is investigating the Fc-enhanced anti-CTLA4 antibody botensilimab in combination with the anti-PD1 antibody balstilimab [105]. The sarcoma cohort included 41 patients, including 16 with LMS (13 STLMS, three ULMS). ORR was 20%, clinical benefit rate was 27%, and 6-month PFS was 40%. Two of the sixteen LMS patients (12.5%) had a partial response. A sarcoma expansion cohort is currently enrolling. Combining chemotherapy with immunotherapy is also being investigated, though with limited success to date. The GEMMK trial tested the combination of gemcitabine plus pembrolizumab and enrolled 19 patients with LMS, and only one patient had a partial response (ORR 5%) [106,107]. A phase II trial of eribulin plus pembrolizumab enrolled 19 patients with LMS (8 STLMS, 11 ULMS) and found one partial response (ORR 5.3%) with a mPFS of 11.1 weeks [108]. Next, the ImmunoSarc2 trial investigated doxorubicin and dacarbazine plus nivolumab for first-line treatment of advanced LMS [109]. The ORR was 56.2% (9 of 16 patients with partial responses), and mPFS was 8.7 months. Another trial is investigating the combination of gemcitabine and docetaxel with the anti-PD1 antibody retifanlimab in advanced soft tissue sarcoma [110]. Ten patients with LMS were enrolled, and three have had partial responses (ORR 33%). The NitraSarc trial is testing the combination of nivolumab and trabectedin in advanced sarcomas [111]. Twenty-eight patients with LMS were enrolled (representing 63%), and PFS at 6 months for the cohort at large (including liposarcomas) was 47.6%. A final trial investigating the combination of chemotherapy and immunotherapy is combining doxorubicin with the anti-CTLA4 antibody zalifrelimab and the anti-PD1 antibody balstilimab [112]. One of the nine patients with LMS has had a partial response, and the trial is ongoing. Lastly, immunotherapy was combined with targeted therapies. For example, a single-arm phase 2 trial investigating the combination of pembrolizumab with the VEGF TKI axitinib in advanced sarcomas [113]. The trial enrolled two patients with STLMS and four with ULMS, and one patient with STLMS had a partial response and another patient with STLMS had stable disease for over 6 months before discontinuing for toxicity. Similarly, the combination of the VEGF TKI lenvatinib plus pembrolizumab was investigated in advanced sarcomas, including 10 LMS [114]. Among these 10 LMS patients, there were no responses though 60% had stable disease, and mPFS was 17.9 weeks. Another trial evaluated the TKI cabozantinib in combination with nivolumab and ipilimumab versus cabozantinib alone in advanced sarcoma, which included 54 patients with LMS [115]. LMS was the histology with the most responses. The ORR was 11% (versus 6% for cabozantinib alone), disease control rate was 80% (versus 42%), and mPFS was 5.4 months (versus 3.8 months).

There are several possible explanations for the underwhelming response seen with immunotherapy in LMS to date. First, compared to carcinomas, LMS generally have lower TMB, which is associated with poorer responses to immunotherapy in numerous other cancer types [116,117]. Furthermore, there are mutations and other mechanisms, such as the frequent loss of *PTEN* in LMS, as previously discussed, that promote immune evasion. *PTEN* mutation or deletion was associated with resistance to immunotherapy in metastatic ULMS [92].

CD47 is a surface receptor expressed by tumor cells that engages SIRPα on phagocytes such as macrophages and dendritic cells, inhibiting macrophage phagocytosis (Figure 4A) [118]. CD47 expression was shown in multiple cancer types, including LMS [119]. Preclinical models have shown benefit of anti-CD47 antibodies to restore macrophage recognition and phagocytosis of tumor cells, including LMS cell line xenografts (Figure 4B) [119]. In these xenograft experiments, anti-CD47 treatment prevented both primary tumor growth and the development of metastatic disease [119]. A phase I/II trial of a novel CD47 decoy receptor (TTI-621) in combination with doxorubicin in LMS reported partial responses in five of twenty evaluable patients (ORR 25%) [120,121]. Unfortunately, further development of this agent and other CD47 inhibitors is being halted due to lack of clinical benefit seen across several other cancer types.

There are other macrophage immune checkpoints and pathways that could be exploited for LMS treatment. For example, CSF1 is a macrophage chemoattractant that is expressed by LMS cells and is associated with macrophage infiltration in LMS, so targeting the CSF1 axis may be worth pursuing in LMS [122]. CSF-1R inhibition in tenosynovial giant cell tumor. As CSF1 expression was found to upregulate PDL1 expression in macrophages [123], the CSF-1R blocking antibody emactuzumab was combined with the anti-PDL1 antibody atezolizumab in advanced solid cancers. In the sarcoma cohort, 1 of 17 (ORR 5.9%) had a partial response [124]. Next, the CD39/CD73 axis is an important regulator of innate and adaptive immunity in the tumor micro-environment. Both of these enzymes contribute to the production of adenosine, which activates the abundant adenosine receptors found on macrophages and inhibits phagocytosis by macrophages [125,126]. CD39 and CD73 inhibitors are being explored for the treatment of solid tumors. Lastly, the CCL2–CCR2 axis is important for monocyte chemotaxis into the tumor micro-environment that is being exploited for treatment in other cancer types. Several different cell types, such as tumor cells and endothelial cells, secrete CCL2, which binds to the CCR2 receptor on monocytes, thereby promoting recruitment of monocytes and their differentiation into macrophages. Inhibitors of this interaction are being investigated in solid tumors [127].

One potential treatment strategy to overcome the resistance caused by the *PTEN* mutation would be to combine immunotherapy, such as anti-CD47, with an AKT/mTOR pathway inhibitor. A potential problem with this strategy is that existing mTOR pathway inhibitors, such as everolimus, inhibit the mTOR pathway in T-cells and can be immunosuppressive [128]. However, the PI3Kβ isoform can regulate AKT activity in tumors with *PTEN* loss and is dispensable for activation of the TCR signaling pathway [129,130]. Additional work will be needed to optimize the targeting of this pathway while preserving anti-tumor immunity.

Future work trying to target the LMS micro-environment will focus on several areas. First, given the macrophage predominance in the LMS micro-environment, more efforts are needed to target this macrophage axis. Targeting macrophages for cancer therapy represents a double-edged sword; on one hand macrophages can promote tumor progression and suppress immune recognition, while on the other hand they can phagocytose cancer cells [131]. Next, different combinations of immunotherapy can be trialed in an effort to enhance anti-tumor immunity [6]. For example, it would be interesting to combine macrophage-targeting drugs (e.g., anti-CD47 or anti-CSF-1R) and T-cell immune checkpoint inhibitors (e.g., anti-PD1) to assess for synergy, as macrophage phagocytosis and subsequent antigen presentation to T-cells can augment T-cell-mediated tumor immunity [132].

## 9. The Leiomyosarcoma Epigenome

Epigenetics refers to changes in the chemical structure of chromatin that do not involve changes in the canonical nucleotide sequence. This can involve alteration of the DNA, such as DNA methylation that often reduces transcription, or post-translational modifications of the histone proteins around which DNA is wrapped. These changes modify the openness of local chromatin regions to regulate gene expression. There is a growing recognition that sarcomas such as LMS are an epigenetically-driven disease [133].

ATRX is a component of the SWI/SNF chromatin remodeling complex that has many different functions and can regulate the epigenome in several different ways. *ATRX* is one of the most commonly altered genes in LMS, in which loss-of-function mutations are seen in ~20% of STLMS and ~30% of ULMS (Figure 1A) [20,21,23,134]. *ATRX* is involved in depositing histone H3 variant 3.3 (H3.3) into the genome at areas of heterochromatin and pericentromeric regions, which helps to maintain these chromatin regions in a compact or silenced state [135,136]. Loss of *ATRX* was shown to increase G-quadruplexes, which are secondary DNA structures found at GC-rich regions of the DNA and are implicated in transcriptional dysregulation and DNA damage, as the presence of G-quadruplexes results in stalled replication forks and DNA damage [137]. *ATRX* is also seen mutated in gliomas but rarely in other cancer types, suggesting that the effects of *ATRX* loss are different based on the cell/tissue of origin. *ATRX* knockout in preclinical glioma models was associated with sensitivity to DNA damaging agents, including PARP and ATR inhibitors [138]. *ATRX* mutation is also associated with increased response to radiation and the oncolytic herpes virus treatment talimogene laherparepvec (TVEC) in a sarcoma cell line xenograft model [139]. *ATRX* may also serve as a link between the epigenome and the host immune response. *ATRX* loss results in impaired cGAS/STING signaling, which is cytosolic DNA-sensing machinery that links the presence of cytosolic DNA (e.g., from viruses and microbes or intrinsic DNA damage) to activation of the innate immune system [139]. *ATRX* mutation is also associated with reduced mast cell infiltration in a xenograft model of sarcoma [140].

Beyond *ATRX*, other epigenetic modifiers are frequently altered in LMS. *MED12* is a component of the kinase module of the mediator complex, which is an epigenetic complex that regulates genome organization and gene expression [141,142,143]. *MED12* mutations occur in 10–20% of LMS and disrupt MED12 regulation of the kinase CDK8 [144,145]. Relatedly, high *CDK8* expression is associated with poor prognosis in ULMS [146]. *MED12* mutation is also seen in approximately 70% of benign leiomyomas [141], suggesting it is likely an early event in uterine leiomyosarcomagenesis. Mechanistic experiments have demonstrated that *MED12* mutation causes changes in the 3D structure of the chromatin in uterine smooth muscle cells, resulting in an aberrant gene expression pattern characterized by increased collagen synthesis and altered tryptophan/kynurenine metabolism [147]. Further, *MED12* mutation disrupts DNA replication, and cells spend longer time in S phase, and are sensitized to DNA-damaging agents such as carboplatin [147]. Other work has shown increased replication stress in *MED12*-mutant cells [148] and associated structural genomic changes and genomic instability [142]. These results suggest the need for investigation of specific DDR inhibitors in *MED12*-mutant LMS.

*MYOCD* encodes myocardin, which is a transcriptional coactivator of the serum response factor (SRF) transcription factor that regulates smooth muscle proliferation [149,150,151]. *MYOCD* is frequently amplified in LMS [16,20,24,152]. *MYOCD* depletion in an LMS cell line reduced smooth muscle differentiation and migration, and overexpression of *MYOCD* in undifferentiated sarcoma cell lines increased smooth muscle differentiation and migration, suggesting MYOCD as a potential therapeutic target in LMS [152], particularly well-differentiated LMS.

*NCOR1* encodes a transcriptional corepressor that regulates transcription factors specific to mesenchymal lineages and can suppress differentiation when overexpressed. Mechanistically, NCOR1 interacts with nuclear hormone receptors such as PPARγ and liver X receptor (LXR) to regulate the expression of metabolism genes. Further, NCOR1 interacts with the histone deacetylase HDAC3, and the deacetylase activity of HDAC3 is dependent on NCOR1 [153]. *NCOR1* amplification occurs frequently in LMS (19% of STMLS and 10% of ULMS in one report) [36]. The activity of NCOR1 is modulated by PI3K/Akt-mediated control of nuclear localization. Therefore, both inhibition of this pathway and of HDAC3 warrant further exploration as potential therapeutic strategies in *NCOR1*-amplified LMS.

DNA methylation is perhaps the most studied epigenetic alteration. DNA methylation profiling of LMS has shown that STLMS and ULMS have different methylation signatures [16,154]. Other DNA methylation studies in sarcoma have shown that analysis of methylation patterns are capable of distinguishing different sarcoma subtypes [155,156]. There has been limited use of DNA hypomethylating agents in sarcoma general, including LMS. One such trial combined the hypomethylating agent decitabine with gemcitabine in sarcomas, finding clinical benefit rate of 58%, and some partial responses were noted in LMS [157]. This may represent a potential future treatment strategy in LMS.

## 10. Telomere Biology

Cancer cells achieve replicative immortality by maintaining their telomeres, the repetitive DNA elements on the ends of chromosomes that normally shorten with each cell division and limit the lifespan of cells [158]. Most normal somatic cells, except stem cells, lack the ability to do this. There are two main mechanisms by which cancer cells maintain their telomeres: TERT (telomerase reverse transcriptase enzyme) expression (85%) or alternative lengthening of telomeres (ALT; 15%) [159]. However, ALT is more prevalent in sarcomas than other types of cancer, and it was reported in 53–78% of LMS, 77% of UPS, and 47–66% of osteosarcomas [20,160,161,162,163,164]. One of the mechanisms leading to ALT is loss of *ATRX* [165]. As mentioned, about 20–30% of LMS harbor *ATRX* mutations. *ATRX* and its partner protein, *DAXX*, interact and help with incorporating the H3 histone variant H3.3 at heterochromatic regions, which has previously been demonstrated to limit ALT [166,167]. C circles (extrachromosomal telomeric repeats) are hallmarks of ALT and are often used to assay for ALT [168,169]. C circles were seen in 38 of 49 (77.6%) of LMS samples in one study [20]. In a meta-analysis of 551 sarcoma patients (226 with ALT and 325 without ALT), including 68 LMS patients, the presence of ALT was associated with higher mitotic count, grade, and worse OS [170].

Targeting telomerase has been a challenge to date. Trials with drugs such as the telomerase inhibitor, imetelstat, were limited by significant hematologic toxicity, limited or no efficacy, and potential off-target effects [171]. Some studies have suggested that ATR inhibitors may be particularly efficacious in the setting of ALT [172]. However, this was not seen in a panel of eight sarcoma cell lines, four each with positive and negative ALT [173], or in the additional cell lines tested [174]. Additional preclinical work is needed to identify vulnerabilities in ALT-positive sarcoma.

## 11. Conclusions and Future Directions

There is great unmet need to improve the systemic treatment of LMS. This review has highlighted both the current state of our molecular understanding of this heterogeneous disease, as well as the potential avenues for targeted therapies (Figure 5). Compared to some cancers with recurrent point mutations conferring sensitivity to drugs, such as the *BRAF* V600E mutation conferring sensitivity to BRAF/MEK inhibition in melanoma, the LMS genome is charactered by low TMB and rare actionable alterations to confer meaningful clinical benefit derived from a targeted therapy. One potential area of future investigation involves utilizing the three transcriptomic subtypes of LMS that have been independently identified across multiple studies and identifying therapies that can be matched to each subtype. This will require rigorously defining the three subtypes and require the development of tests in the clinic to delineate these three subtypes in order for subsequent clinical trials to tailor subtype-specific therapies. Additional preclinical work is also needed to identify novel treatments that are specific for each subtype. Next, targeting the macrophage-rich micro-environment is an interesting area of further investigation. Immune checkpoint blockade has been largely ineffective in LMS to date, and perhaps one of the reasons is the lack of T-cell infiltration in the LMS micro-environment, which may be in part explained by immunosuppressive M2 macrophage enrichment. The novel combination of immunotherapies also represents an interesting area of future investigation. Lastly, given the recurrently reported alterations in the DDR pathway in LMS, targeting this pathway for LMS treatment is of great interest. Trials are ongoing to test agents such as PARP inhibitors in LMS. It is likely that a combination of treatments targeting multiple vulnerabilities may be needed in LMS. In developing and implementing novel targeted therapies for LMS treatment, it will be important to consider potential toxicity relative to that of currently used medicines and the cost effectiveness of implementing these novel treatments.

## Figures and Tables

**Figure 1 cancers-16-00938-f001:**
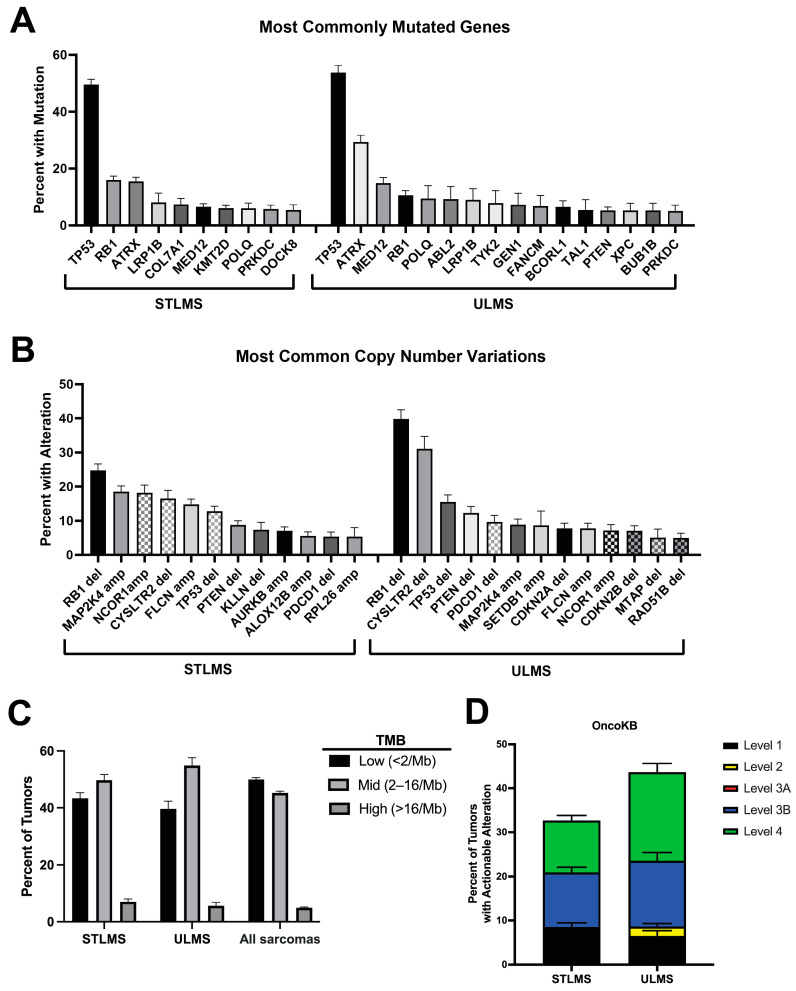
Clinically actionable alterations in leiomyosarcoma from the GENIE database. AACR GENIE data (version 14.1) were queried using cBioPortal [37,38]. For STLMS, 723 tumor specimens from 699 patients. For ULMS, 374 tumor specimens from 359 patients. Genes were only included if they were profiled in at least 10% of tumors. Genes are displayed on the graph if they were altered in at least 5%. (**A**) Most commonly mutated genes. (**B**) Most common copy number alterations. (**C**) Tumor mutational burden, divided by low (<2 mutations per megabase), medium (2–16), or high (>16). (**D**) Percent with clinically actionable alterations by OncoKB level. A Level 1 alteration is an FDA-recognized biomarker predictive of response to an FDA-approved drug. A Level 2 alteration is a biomarker recommended by the National Comprehensive Cancer Network (NCCN) or other professional organizations that predict response to an FDA-approved drug. A Level 3A alteration describes compelling clinical evidence to support that the alteration predicts response to a drug. A Level 3B alteration describes an alteration for which there is an FDA-approved or investigational drug for another indication (usually a different tumor type). A Level 4 alteration has compelling biological evidence that it predicts response to a drug. Throughout the figure, bars represent percentages plus standard error of proportion.

**Figure 2 cancers-16-00938-f002:**
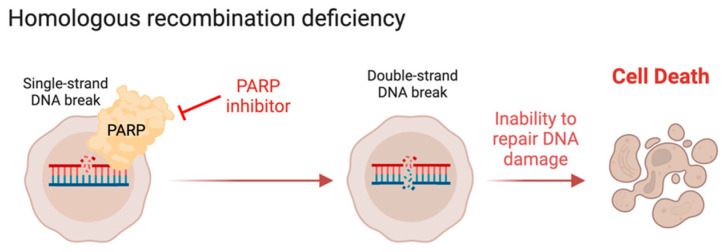
Homologous recombination deficiency sensitizes to PARP inhibition. Homologous recombination (HR) deficiency can result from loss of HR genes such as BRCA1 and BRCA2, which creates synthetic lethality with PARP inhibition. This may be a strategy that can be implemented to treat LMS with HR deficiency. Created with BioRender.com (accessed on 2 February 2024).

**Figure 3 cancers-16-00938-f003:**
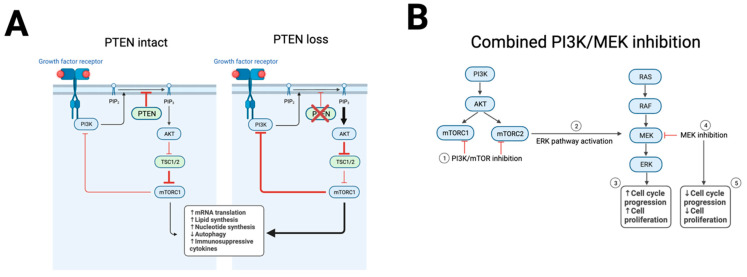
PTEN loss is common in LMS. (**A**) With intact *PTEN*, the PI3K pathway is inhibited. With *PTEN* loss, the PI3K pathway is activated, resulting in downstream pathway activation. (**B**) Combined PI3K and mTOR inhibition was shown to result in compensatory ERK pathway activation, increasing cell proliferation. A possible combination to treat LMS is with dual PI3K/mTOR and ERK pathway inhibitors. Created with BioRender.com (accessed on 2 January 2024).

**Figure 4 cancers-16-00938-f004:**
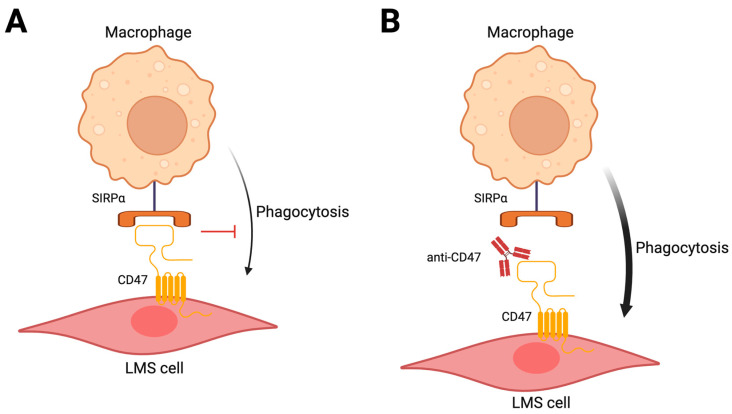
Targeting the CD47–SIRPα axis to treat leiomyosarcoma. (**A**) Binding of SIRPα on macrophages to CD47 on LMS cells inhibits macrophage phagocytosis. (**B**) Therapeutic inhibition of CD47 promotes macrophage phagocytosis. Created with BioRender.com (accessed on 2 January 2024).

**Figure 5 cancers-16-00938-f005:**
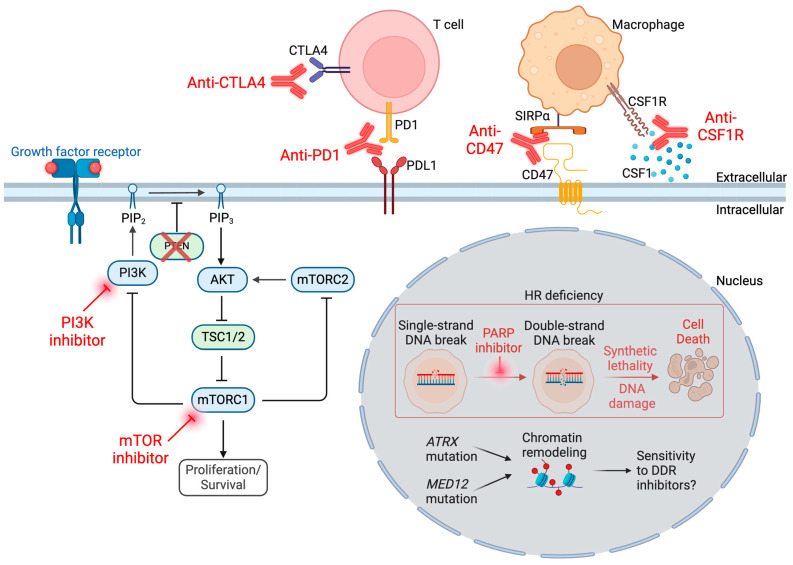
Summary of current therapies being investigated in LMS. Created with BioRender.com (accessed 16 February 2024).

**Table 1 cancers-16-00938-t001:** Summary of genomic studies in leiomyosarcoma.

Ref.	Modality	Total n	STLMS	ULMS	Major Findings	*TP53*	*RB1*	*ATRX*	*CDKN2A*	*CDKN2B*	*PTEN*	Other
[16]	WES	80	53	27	Predominantly chromosomal or arm-level deletions.	Mutation, 50%; deep deletion, 9%	Deep deletion, 14%		Deep deletion, 14%		Deep deletion, 13%	
[17]	Targeted exome (230 genes)	35	35	0	Losses of chromosomal regions involving key tumor suppressor genes *PTEN* (10q), *RB1* (13q), *CDH1* (16q), and *TP53* (17p) were the most frequent genetic events. Gains mainly involved chromosome regions 17p11.2 (*MYOCD*) and 15q25-26 (*IGF1R*).	Mutation, 37%; deletion, 43%	Mutation, 8.5%; deletion, 54%				Deletion, 60%	*CDH1* deletion, 46%
[18]	Targeted exome (47 genes)	751	350	401	*TP53* mutations in 42.2% of STLMS and 40.5% of ULMS, *BRCA2* mutations in 11% of STLMS and 21.7% of ULMS. *PTEN* mutations in 6.3% of STLMS and 0% of ULMS.	Mutation, 41.7%	Mutation, 5.3%				Mutation, 4.4%	*BRCA2* mutation, 17.1%
[19]	SNP arrays, RNA seq, WGS on subset	84	0	84	Alterations affecting *TP53*, *RB1*, *PTEN*, *MED12*, *YWHAE* and *VIPR2* were present in the majority of ULMS. Enrichment in PI3K/AKT/mTOR, estrogen-mediated S phase entry, and DNA damage response signaling pathways.	Altered in 92%; mutation, 41.7%; deletion, 33%	88%				75%	Mutations in *MED12*, 12.5%; *EIF3A*, 16.7%; *ABL1*, 12.5%, *IGF2R*, 12.5%; *ATR*, 8.3%; *RAD50*, 8.3%. *BRCA1*, 8.3%.
[20]	WES	49	39	10	Notable mutational heterogeneity, near-universal loss of *TP53* and *RB1*, widespread DNA copy number alterations with evidence of chromothripsis, and frequent whole-genome duplication.	49%	27%	24%			57%	
[21]	WES	19	0	19	*TP53*, *MED12*, and *ATRX* mutations were prevalent.	33%	26%					*MED12*, 21%
[22]	Targeted exome (151 genes)	25	16	9	CNVs were identified in 85% of cases. Most frequent losses in chromosomes 10 and 13 including *PTEN* and *RB1*. Most frequent gains in chromosomes 7 and 17.	36%	12%	16%				*ATM*, 16%; *EGFR*, 12%
[23]	Targeted exome (341–468 genes)	80	0	80	Compared to ESS, STUMP. *PTEN* alteration frequency was higher in the metastases samples as compared with the primary samples. Genomes of low-grade tumors were largely silent, while 50.5% of high-grade tumors had whole-genome duplication.	56%	51%	31%				
[24]	WGS	53 samples (34 patients)	23	11	Mutational signatures highlight importance of DNA damage repair and homologous recombination deficiency. Dystrophin deletion associated with worse outcome. Whole-genome doubling was prevalent. Analysis of matched primary-metastatic samples suggested divergence 10–30 years prior to diagnosis.	Mutation, 82.3%; deletion, 14.7%	Mutation, 11.8%; deletion, 8.8%	Deletion, 8.8%				

WES, whole exome sequencing. WGS, whole genome sequencing. SNP, small nucleotide polymorphism.

**Table 2 cancers-16-00938-t002:** Summary of transcriptomic studies in leiomyosarcoma.

Ref.	Total n (LMS)	STLMS	ULMS	Modality	Major Findings
[16]	80	53	27	Bulk RNA sequencing	Identified three subgroups: ULMS group with poor prognosis and two STLMS clusters (C1, C2). C1 with hypermethylation, higher expression of *IGF1R* and cell cycle control genes, DNA replication, DNA repair, *RB1* mutations, *PTEN* deletion. C2 with more inflammatory cells (NK and mast).
[20]	49	39	10	Bulk RNA sequencing	Identified three subgroups. Subgroup 1 with platelet degranulation, complement activation, and metabolism signatures. Subgroup 2 with muscle development and regulation of membrane potential signatures. Subgroup 3 with myofibril assembly, muscle filament function, and cell–cell signaling signatures.
[40]	51	35	16	Microarray	Identified three subgroups. Subgroup 1 with muscle contraction and actin cytoskeleton genes. Subgroup 2 with protein metabolism, cell proliferation, and organ development genes. Subgroup 3 with CSF1 response genes.
[42]	99	50	49	Bulk RNA sequencing	Identified three subtypes. Validated their findings from Beck et al. study using new cohort (*n* = 99) and TCGA data (*n* = 82). Identified IHC-compatible assays for different STLMS subtypes.
[19]	24	0	24	Bulk RNA sequencing	Enrichment in PI3K/AKT/mTOR, estrogen-mediated S phase entry, and DNA damage response signaling pathways.
[39]	40			Microarray	Identified a muscle gene-enriched group of 11. Remaining 29 were heterogeneous.
[41]	17			Microarray	No difference among anatomic site, tumor grade, or metastatic lesions. ULMS enriched for site-specific genes such as regulators of urogenital differentiation, development, and growth (*ESR1*, *HOXA10*, *PBX1*, and *FAT*) compared to STLMS.
[24]	113 (130 samples, 51 newly sequenced, 79 from TCGA)	23	11	Bulk RNA sequencing	Identified three subtypes. Subtype 1 contained LMS from different anatomic sites, harbored higher TMB, and was associated with worse OS and DSS. Subtype 2 was mostly abdominal and was associated with better OS and DSS compared to the other subtypes. Subtype 3 was mostly uterine, harbored higher TMB, and was associated with worse OS and DSS. Matching primary-metastatic samples allowed for assessing tumor evolution; metastases maintained subtype.

## Data Availability

Data from the GENIE database are available through cBioPortal. Our analyses can be shared upon request to the corresponding author.

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
