# Peer review of "The Future of Targeted Therapy for Leiomyosarcoma"

_cancers, 2024, doi:10.3390/cancers16050938_

Round 1
Reviewer 1 Report
Comments and Suggestions for Authors
The authors have to be commended for a comprehensive review of current and potential future therapies for LMS, which is highly relevant to the field of oncology. and sarcoma research. It thereby provides a valuable resource for researchers and clinicians in the field.
There are the following suggestions to be considered:
-the gap between research findings and clinical application remains a point for improvement. This is particularly important considering the wealth of information available but without any major direct help for the patient.
-the review lists future potential avenues, but it's more a listing of technical aspects to pursue, no judging of what also makes sense given the investments already done. Respecting all the potential benefits of these new avenues, the potential sides effects of such novel therapies are always neglected. This needs to be incorporated, as well as cost efficiency analyses.
-this review is very complex. It would greatly help to include kind of a summarising figure, so that the reader has an overview what was discussed.
Author Response
The authors have to be commended for a comprehensive review of current and potential future therapies for LMS, which is highly relevant to the field of oncology. and sarcoma research. It thereby provides a valuable resource for researchers and clinicians in the field.
We thank the reviewer for this kind feedback.
There are the following suggestions to be considered:
-the gap between research findings and clinical application remains a point for improvement. This is particularly important considering the wealth of information available but without any major direct help for the patient.
-the review lists future potential avenues, but it's more a listing of technical aspects to pursue, no judging of what also makes sense given the investments already done. Respecting all the potential benefits of these new avenues, the potential sides effects of such novel therapies are always neglected. This needs to be incorporated, as well as cost efficiency analyses.
The reviewer brings up great points. We have tried to emphasize the areas of future application to clinical practice that we think are most exciting in the final section entitled “Future Directions.” It is difficult to comment on the potential side effects and cost effectiveness in the case of targets for which drugs are not yet available or need to be tested in patients with LMS. However, we do mention the importance of considering potential side effects and cost effectiveness analyses, which was added to the “Future Directions” section at the end of the manuscript as follows: “In developing and implementing novel targeted therapies for LMS treatment, it will be important to consider potential toxicity relative to that of currently used medicines and the cost effectiveness of implementing these novel treatments.”
-this review is very complex. It would greatly help to include kind of a summarising figure, so that the reader has an overview what was discussed.
This is great suggestion and we thank the reviewer for it. We have created a new Figure 5 that summarizes potential areas of therapeutic investigation moving forward and believe this strengthens our manuscript.
Reviewer 2 Report
Comments and Suggestions for Authors
The manuscript entitled “The Future of Targeted Therapy for Leiomyosarcoma” by Denu R. et al aims to review several molecular profiling studies in leiomyosarcoma histotype, highlighting hopeful approaches of current research. The review very well suggests possible future strategies to improve the use of molecularly-matched treatment in LMS. including targeting the DNA damage response, the macrophage-rich micro-environment, the PI3K/mTOR pathway, epigenetic regulators, and telomere biology.
The review is well written and the search strategy and the study selection process are well organized. The manuscript has the potential to be accepted, but some minor points have to be fixed before.
Minor revisions:
1. In the text body Table 2 is not mentioned. The authors could include Table 2 in paragraph 3 line 154. Also, could you include the Table 2 after this part?
2. Table 1 and Table 2 are very exhaustive, but they should be ameliorated including the reference numbers instated of First author and PMID columns, making it more readable. Also, the table style should be improved or should be written less in the columns.
3. I recommend that the authors include also a small comment about the attraction of pharmacogenomic studies in leiomyosarcoma treatments.
Author Response
The manuscript entitled “The Future of Targeted Therapy for Leiomyosarcoma” by Denu R. et al aims to review several molecular profiling studies in leiomyosarcoma histotype, highlighting hopeful approaches of current research. The review very well suggests possible future strategies to improve the use of molecularly-matched treatment in LMS. including targeting the DNA damage response, the macrophage-rich micro-environment, the PI3K/mTOR pathway, epigenetic regulators, and telomere biology.
The review is well written and the search strategy and the study selection process are well organized. The manuscript has the potential to be accepted, but some minor points have to be fixed before.
We thank the reviewer for this kind feedback and their very good suggestions to improve our manuscript.
Minor revisions:
- In the text body Table 2 is not mentioned. The authors could include Table 2 in paragraph 3 line 154. Also, could you include the Table 2 after this part?
We thank the reviewer for catching this error, and we have now mentioned Table 2 in the body of the text.
- Table 1 and Table 2 are very exhaustive, but they should be ameliorated including the reference numbers instated of First author and PMID columns, making it more readable. Also, the table style should be improved or should be written less in the columns.
We thank the reviewer for this suggestion to improve the readability of the tables, and we have made the recommended corrections.
- I recommend that the authors include also a small comment about the attraction of pharmacogenomic studies in leiomyosarcoma treatments.
We thank the reviewer for this suggestion. We have added a section about pharmacogenomic biomarkers in LMS on page 10.